# Crustal melting in orogenic belts revealed by eclogite thermal properties

**Baohua Zhang** [1] ✉, **Hongzhan Fei** [2], **Jianhua Ge**[3,4], **Lingsen Zeng**[5] & **Qunke Xia** [1] ✉

Partial melting in the continental crust may play a critical role on the behavior of continents during collision. However, the occurrence of partial melt in orogenic continental crust is not well understood. Since the temperature of the orogen is controlled by the thermal properties of constituent rocks, we measured the thermal conductivity and diffusivity of eclogite, the most important ultrahigh pressure metamorphic rocks, as a function of pressure, temperature, composition, and water content, and simulated the thermal structure of the Sulu and Himalaya-Tibet orogens in eastern and southwestern China, respectively. Our results show that the temperature at ~30-km depth beneath the orogens reaches the solidus of wet granite and phengite (~940 K), therefore, the partial melting in the orogenic continental crust is well explained. The melt may facilitate the exhumation of subducted crust, produce the low seismic-velocity zone, and cause the high-conductivity anomaly in the shallow depth of orogenic belts.

The most outstanding geophysical feature of the crust of orogenic belts is the presence of a widespread seismic low-velocity zone at depths of 20–50 km in global collision regions such as western Norway[1], Himalaya-Tibet in southwestern China[2,3], and Sulu in eastern China[4]. The seismic low-velocity zone is also frequently associated with high electrical conductivity values (up to 0.1 S/m), as interpreted by magnetotelluric (MT) observations[5–9]. The origin of this low seismic velocity and high electrical conductivity remains highly debated. A few studies attributed these features to the hydration of eclogite[10], which is composed of omphacite and garnet and is the main constituent rock of subduction zones and in the deep part of thickened continental crust[1,3,11–13]. However, this hypothesis was later disproven by electrical conductivity experiments, which showed that hydrous eclogite cannot produce a conductivity value of 0.1 S/m[14]. Instead, partial melting can explain the observed geophysical anomalies in orogenic belts[2,5–8] because the presence of melt can reduce the seismic velocity[15] and enhance the electrical conductivity[16–18].

The question is why the orogenic continental crust can partially melt. As it is known, the melting of lithospheric rocks is controlled by the temperature distribution of the lithosphere, which is governed by heat flow from the deep mantle to the surface and thus controlled by the thermal properties of rocks and minerals, i.e., thermal diffusivity ($D$) and thermal conductivity ($\kappa$). A comprehensive understanding of the $D$ and $\kappa$ of eclogite under lowermost continental crust conditions is therefore necessary.

Nevertheless, the $D$ and $\kappa$ of eclogite under relevant high-pressure and high-temperature conditions of deep crust remain poorly constrained owing to experimental difficulties. A previous study reported $D$ in a synthetic eclogite sample with a single component (MORB-composition) at 3.5 GPa[19], and $\kappa$ was calculated from $D$[20] indirectly. However, the effects of pressure and modal mineralogy on the $D$ and $\kappa$ values of eclogite remain unknown, whereas such information is indispensable to infer the thermal structure of the continental lithosphere and thus to interpret the generation of

[1]Key Laboratory of Geoscience Big Data and Deep Resource of Zhejiang Province, School of Earth Sciences, Zhejiang University, Hangzhou 310027, China. [2]Bayerisches Geoinstitut, University of Bayreuth, Bayreuth D-95440, Germany. [3]Advanced Batteries and Materials Engineering Research Center, Guizhou Light Industry Technical College, Guiyang 550025, China. [4]Key Laboratory for High-Temperature and High-Pressure Study of the Earth's Interior, Institute of Geochemistry, Chinese Academy of Sciences, Guiyang 550081, China. [5]Institute of Geology, Chinese Academy of Geological Sciences, Beijing 100037, China. ✉e-mail: zhangbaohua@zju.edu.cn; qkxia@zju.edu.cn

partial melting and metamorphic and geochemical geodynamics of collisional orogen.

On the other hand, eclogites are formed through high-pressure to ultrahigh-pressure metamorphism. Regardless of the protoliths (e.g., gabbros or basaltic rocks), eclogites usually contain water in the constituted hydrous minerals (e.g., phengite, lawsonite, epidote) and nominally anhydrous minerals (e.g., omphacite, garnet). Upon with the increasing of temperature during continental deep subduction, the breakdown of hydrous minerals would leave omphacite and garnet as the main water reservoir in eclogite[21–23]. Because water may affect the thermal properties of minerals[24,25], it is also necessary to investigate the water content dependence of $D$ and $\kappa$.

In this study, multi-anvil experiments were performed to measure $D$ and $\kappa$ simultaneously in natural eclogite with various modal mineralogy and water contents at temperatures of 300–873 K and pressures of 1–3 GPa, corresponding to crust and lithospheric mantle conditions. Eclogite samples were collected from the Dabie-Sulu ultrahigh-pressure metamorphic (UHPM) belt in eastern China, which is the largest UHPM belt in the world. This belt is separated into western (Dabie UHPM belt) and eastern (Sulu UHPM belt) sections by the northeast-southwest trending Tan-Lu fault, thus two samples collected from representative outcrops in the Dabie UHPM belt (DB11 and DB13) and the other two samples from the Sulu UHPM belt (SL9 and SL12) were used (Supplementary Fig. 1).

## Results and discussion

### Temperature and pressure dependences of $D$ and $\kappa$ in eclogite

The $D$ and $\kappa$ values obtained under the investigated pressure and temperature conditions are listed in Supplementary Table 1. At 3 GPa, both $D$ and $\kappa$ are found to systematically decrease by a factor of 2–3 with increasing temperature from 300 to 873 K (Fig. 1a, b), and the temperature dependence decreases with increasing temperature. Despite the limited pressure range (1–3 GPa), Fig. 1c, d clearly show that both $D$ and $\kappa$ increases linearly with increasing pressure at given temperature conditions (300 and 823 K), whereas the pressure derivative ($\partial D/\partial P$ and $\partial \kappa/\partial P$) is steeper at 300 K than 823 K.

The measured $D$ and $\kappa$ for each sample were fitted by a power-law formulation[26] to simultaneously consider the influence of pressure and temperature on the thermal transport properties:

$$D = D_0 \left(\frac{300}{T}\right)^{n_D} (1 + aP) \tag{1}$$

$$k = k_0 \left(\frac{300}{T}\right)^{n_k} (1 + bP) \tag{2}$$

where $D_0$ and $\kappa_0$ are the fitting results of $D$ and $\kappa$ under ambient conditions, respectively, $P$ is pressure (GPa), $T$ is temperature (K), $n_D$ and $n_\kappa$ are the temperature exponents, and $a$ and $b$ are the pressure

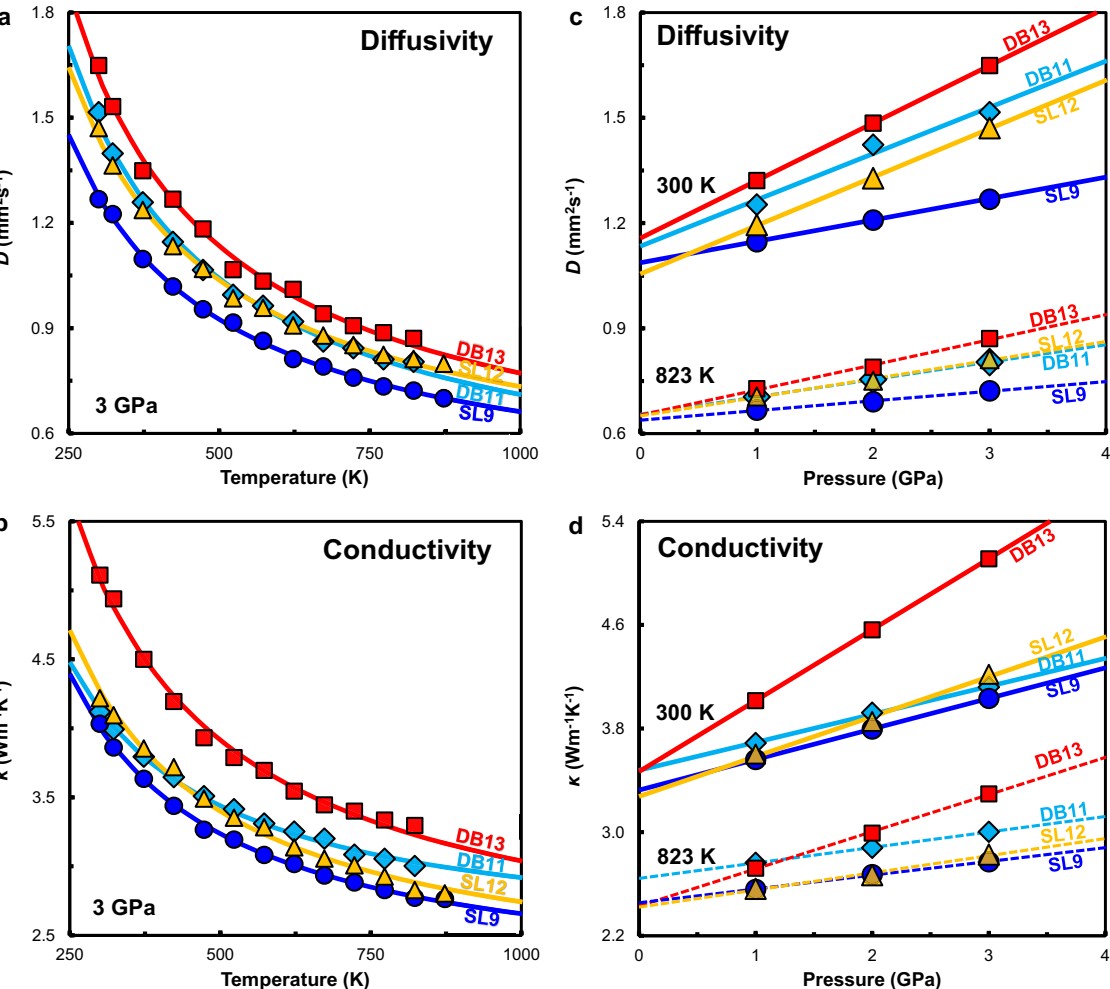

**Fig. 1 | Temperature and pressure dependences of the thermal diffusivity ($D$) and conductivity ($\kappa$) of eclogite. a** Temperature dependence of $D$ at 3 GPa. **b** Temperature dependence of $\kappa$ at 3 GPa. **c** Pressure dependence of $D$ at 300 and 823 K. **d** Pressure dependence of $\kappa$ at 300 and 823 K. The experimental error is less than 1.5% (Supplementary Table 1), and the error bars are smaller than the symbols.

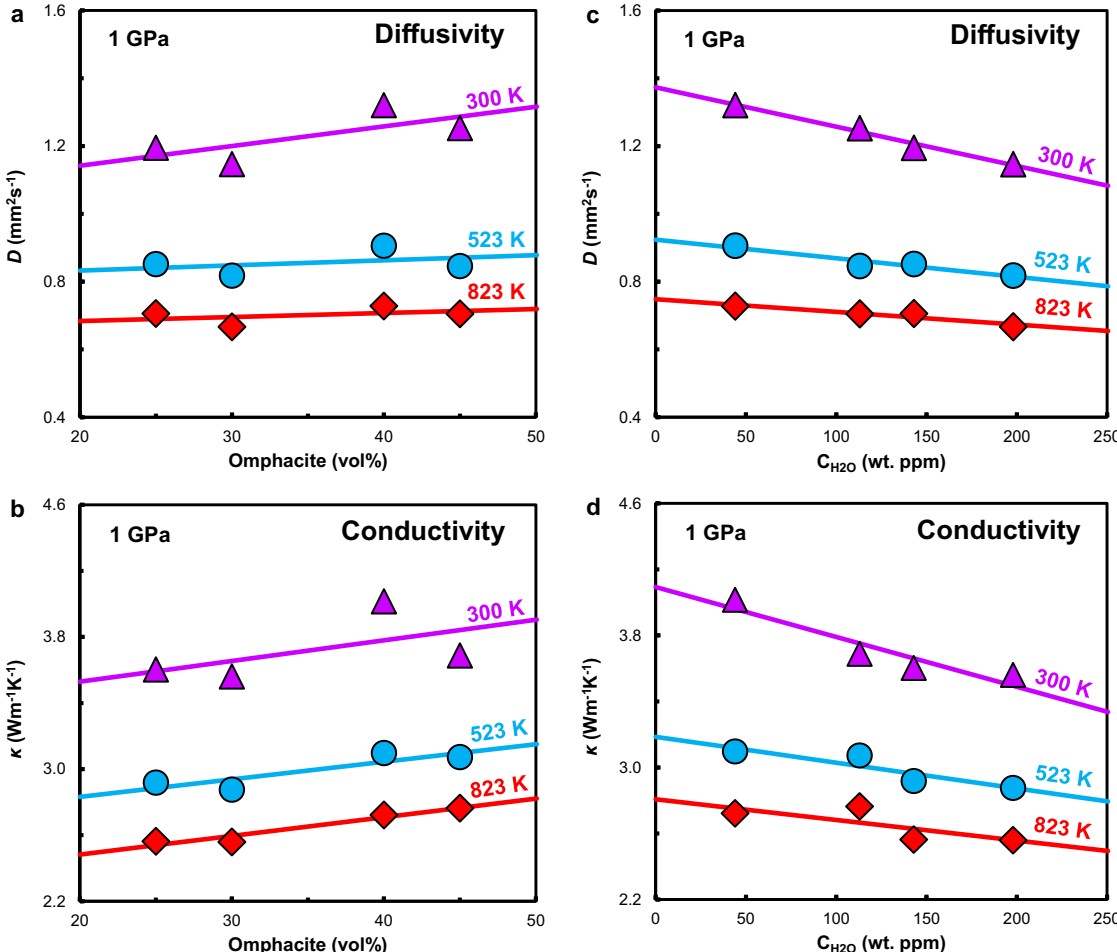

**Fig. 2 | Effects of modal mineralogy and water content on the thermal properties of eclogite at 1 GPa and 300–823 K. a** Effect of omphacite proportion on the thermal diffusivity ($D$). **b** Effect of omphacite proportion on the thermal conductivity ($\kappa$). **c** Effect of water content on $D$. **d** Effect of water content on $\kappa$. The experimental error is less than 1.5% (Supplementary Table 1), and the error bars are smaller than the symbols.

coefficients. The least square fitting yields $n_D$, $n_\kappa$, $a$, and $b$ values of 0.53–0.65, 0.31–0.49, 0.06–0.13 GPa$^{-1}$, and 0.06–0.24 GPa$^{-1}$, respectively (Supplementary Table 2).

Although the absolute values of $D$ and $\kappa$ in the four samples differ by ~30%, nearly identical temperature and pressure dependences were shown in this study. In terms of the heat transport mechanism, heat transfer in materials is known to be controlled by two subatomic processes: (1) the transfer of vibrational energy from one bond to another (i.e., phonon-phonon transport), which is inversely proportional to temperature and known as lattice thermal conductivity ($\kappa_{lat}$); and (2) the conversion of vibrational energy to light (i.e., phonon-photon conversion), which is proportional to the cube of temperature, and is known as radiative thermal conductivity ($\kappa_{rad}$). The $\kappa_{lat}$ and $\kappa_{rad}$ usually dominate at temperatures below and above 1000 K, respectively[27,28]. The continuous decreasing of $D$ and $\kappa$ with increasing temperature up to 873 K in this study suggest that the thermal properties of eclogite is dominated by $\kappa_{lat}$. This is also confirmed by the theoretical calculation (Eq. 4): the calculated $\kappa_{rad}$ is less than 3% of $\kappa_{lat}$ at 300–873 K, namely, $\kappa_{rad}$ has negligible effect to the experimental determination of $\kappa_{lat}$. On the other hand, the increase of $D$ and $\kappa$ with pressure is previously interpreted to be caused by the decrease of phonon scattering probability[29].

### Influence of modal mineralogy and water

Mineral proportions may affect the $D$ and $\kappa$ of the bulk rock[27,28] because different minerals have different thermal properties. In this study, the

eclogite samples were composed of 40–60 vol% garnet, 25–40 vol% omphacite, approximately 2 vol% quartz, and other minor phases including rutile, phengite, symplectite, apatite, epidote, amphibole, zoisite, and kyanite, with concentrations of less than 10 vol% each (Supplementary Fig. 2, Supplementary Table 3). The contributions of quartz and other minor phases should be negligible because of their small proportions even though quartz has higher $D$ and $\kappa$ than garnet and omphacite[20,27,28,30]. Additionally, although the fraction of garnet is higher than that of omphacite, the $D$ and $\kappa$ of eclogite is found to be nearly independent of garnet content (Supplementary Fig. 3a, b) and slightly positively correlated with omphacite content (Fig. 2a, b). This is because the $D$ and $\kappa$ values of garnet are considerably lower than those of omphacite[20,30]. Despite of the higher density and sound velocity of garnet than omphacite, specific heat capacity of omphacite is much larger than that of garnet, which would result in a larger mean free path for omphacite, namely, higher $\kappa$ or $D$ (Method section).

Although the thermal properties of eclogite have been determined experimentally[19,20], the effect of water on the $D$ and $\kappa$ of eclogites remains unknown. In this study, it is demonstrated that the $D$ and $\kappa$ of eclogite decrease linearly with increasing water content (Fig. 2c, d). Since thermal properties of eclogite are primarily controlled by omphacite as discussed above, water-content dependence of $D$ and $\kappa$ in eclogite should be controlled by the water-content in omphacite. In the viewpoint of mineral physics, water in eclogites mainly exists as hydroxyl in the crystal structures of garnet and omphacite[10,14,31,32] (Supplementary Fig. 4). Hydroxyl could distort the crystal structures,

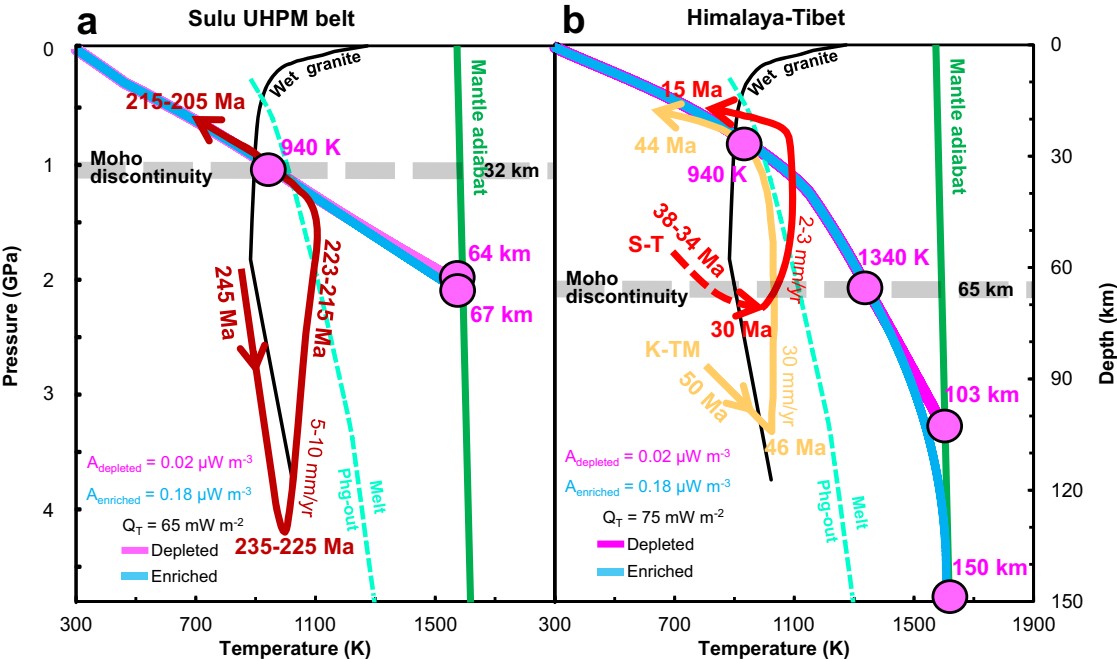

**Fig. 3 | Geotherms of the crust and lithospheric mantle beneath orogenic belts.** **a** Sulu UHPM belt, **b** Himalaya-Tibet. The thick purple and light blue curves indicate the geotherm profiles calculated based on heat generation of 0.02 μWm⁻³ in the depleted mantle and 0.18 μWm⁻³ in the enriched mantle[51], respectively. Solid black line: solidus of wet granite[54]; thick light cyan dashed line: dehydration melting curve of phengite[53]; thick green line: mantle adiabatic line ($T_m = 1573 + 0.3z$ where $z$ is depth). The intersection between the calculated geotherm profile and mantle adiabatic line indicates the thickness of the lithosphere, and the intersection between the geotherm profile and Moho discontinuity indicates the Moho temperature. *P-T-t* paths: The thick dark red line with arrows in **a** represents the Sulu UHPM belt[21, 58, 60]. The thick red and orange lines with arrows in **b** represent the Southern Tibet (S-T)[38, 64] and Kaghan and Tso Morari (K-TM) terranes[63], respectively.

which produces additional phonon vibrational modes arising from new O–H bonds[24,25,33] and thus enhancing the frequency of scattering events. It sequentially reduces the number of free paths for phonons and accordingly reduces $D$ and $k$[24,25].

### Thermal structure and partial melting of orogenic belts

Numerous studies have provided petrological evidences for the occurrence of partial melting in natural UHP rocks during the continental collision[34–40]. However, the temperature distribution beneath continental orogenic belts, which is essential for interpretation of the partial melt process during continental collision, is poorly understood. Based on the $\kappa$ of eclogite under lithospheric pressure, temperature, and water-content conditions determined in this study and that of other rocks and minerals (olivine, granite, gneiss, amphibolite, and granulite) from literatures[25,28,41,42], numerical thermophysical modeling was performed to calculate the geotherms in the lithosphere of the Sulu orogenic belt (normal crust thickness, ~32 km) in eastern China and the Himalaya-Tibet orogenic belt (thickened crust, ~65 km thick) in southwestern China as two typical examples. The crustal structures, composition, and heat production of each belt was given in Supplementary Fig. 5. The water content in the continental mantle lithosphere was assumed to be constant ~100 wt. ppm[43]. The surface heat flow ($Q_T$) was estimated to be 65 mW m⁻² for the Sulu belt and 75 mW m⁻² for the Himalaya-Tibet belt based on the database of surface heat flow[44,45]. The mantle adiabat was set to a potential temperature of 1573 K with a gradient of 0.3 K/km, and the surface temperature was fixed at 283 K. A one-dimensional model with a boot-strapping approach (Eqs. 8–10)[46,47] was used. Details of the calculation is given in the Methods section.

Figure 3 shows the modeled results of the temperature-depth profiles and average lithospheric thickness of the Sulu and Himalaya-Tibet belts. In the Sulu orogenic belt, the Moho temperature at depth of 32 km is ~940 K, and the inferred lithospheric thickness is around 64–67 km (Fig. 3a). In the Himalaya-Tibet orogenic belt, due to the differences in heat flow and heat production (Supplementary Fig. 5),

the temperature at Moho discontinuity (~65 km)[48,49] reaches to 1340 K, and the lithospheric thickness is close to 150 km (Fig. 3b). The effect of radiogenic heat production (depleted versus enriched)[50–52] on the lithospheric thickness is large in the Himalaya-Tibet orogen but negligible in the Sulu orogenic belt. Note that, although the crustal structure, composition, heat flow, and radiogenic heat production in the Sulu belt differ from those in the Himalaya-Tibet belt (Supplementary Fig. 5), the temperature (~940 K) at ~30 km depth of Sulu (base of the crust, close to Moho discontinuity of Sulu belt) is almost identical to that of Himalaya-Tibet belt at the same depth (middle crust of Himalaya-Tibet belt), and close to the solidus of phengite[53] and wet granite[54] (Fig. 3). Therefore, the shallowest regions for the occurrence of partial melt beneath the Sulu and Himalaya-Tibet belts are ~30 km depth, i.e., at base of crust near the Moho discontinuity and the middle crust, respectively. Namely, crustal melting in orogenic belts could occur beneath both the Sulu and the Himalaya-Tibet belts.

In addition to the Sulu and the Himalaya-Tibet belts, the other UHP belts worldwide, such as the Kokchetav massif (Kazakhstan), the Bohemia massif (Central Europe) and the Greenland, the temperature at ~30 km depth could reach the dehydration temperature of hydrous rocks as well[21,22,55]. Water released by the dehydration depresses the melting point of rocks considerably, leading to partial melting in the deeper and hotter regions[21,22,56] (Fig. 3).

Here the questions about when and why partial melting have occurred in the continental crust of the Sulu and the Himalaya-Tibet belts during their subduction and exhumation are discussed in detail. According to the SHRIMP U-Pb analysis of zircon[57–60], there are three important stages of pressure-temperature-time (P-T-t) paths for the metamorphic history of the Dabie-Sulu UHP rocks (Red curves in Fig. 3a), i.e., (1) 246–244 Ma for early-stage quartz eclogite-facies prograde metamorphism at 843–963 K and 1.7–2.1 GPa; (2) 235–225 Ma for peak UHP metamorphism at 1023–1123 K and 3.0–4.3 GPa; (3) 215–205 Ma for late-stage amphibolite-facies retrogression at 823–923 K and 0.7–1.0 GPa. The first episode of partial melting of Sulu

belt should occur at the beginning of the early stage of exhumation because the temperature of the exhumation rocks reaches the solidus of wet granite at ~120 km depth (3.8 GPa, 1050 K) (Fig. 3a), indicated by the occurrence of nanogranites in zircon, garnet and monazite from diatexite[40]. Upon the decompression of "hot" exhumation, the breakdown of phengite could provide sufficient amount of the water and inducing the second episode of partial melting at ~60 km depth (2.0 GPa, 1095 K)[11,21,39,40,57–62], which was recorded by the presence of multiphase solid inclusions within UHP minerals and anatectic zircons and minor interstitial K-feldspar between garnet and omphacite in the Sulu UHP eclogites[37,39,40].

The Himalaya-Tibet belt has two types of eclogite with different P-T-t paths for subduction and exhumation processes[38,63,64]: (1) ~50 Ma UHP eclogites in the Kaghan and Tso Morari massifs in the western Himalaya (K-TM curve in Fig. 3b); (2) ~30 Ma HP eclogites distributed from central Himalaya to Southern Tibet (S-T curve in Fig. 3b). As illustrated in Fig. 3b, K-TM UHP slices has undergone one episode of partial melting at ~100 km depth (3.2 GPa, 1025 K) due to melting of fertile components at the initial stage of exhumation, whereas fluid-absent melting of phengite may continuously occur during the fast exhumation of felsic UHP rocks[62]. In contrast, the S-T UHP slice have undergone the first episode at ~70 km depth (2.1 GPa, 900 K, near Moho discontinuity of Himalaya-Tibet belt) due to the melting of fertile components and the second episode of partial melting at ~55 km depth (1.6 GPa, 1080 K) due to the melting of phengite.

The partial melt plays important roles in geodynamic and geophysical processes. For example, when partial melting takes place at the peak UHP metamorphic stage or at the final stage of subduction, the melts can reduce the strength the rocks, leading to the weakening of the deeply subducted continental crust and the decoupling of rocks from the subducting slab, which may result in the initiation of exhumation[65–67]. When partial melting takes place during exhumation (generally caused by the breakdown of hydrous UHP minerals and the exsolution of water in NAMs), the melt channel may lubricate the edge of exhumated slices of UHP rocks and thus enhance the transport of materials from deep regions to the surface[11,21–23,39,67].

The onset of partial melting during subduction and exhumation of continental crust should also cause geophysical anomalies because partial melt could lower the seismic velocity and enhance the electrical conductivity of rocks[15–18]. Therefore, the occurrence of partial melting in the lithosphere could explain the low seismic-velocity and high electrical conductivity observed in the lithosphere of Sulu[4,8] and Himalaya-Tibet belts[2,3,5–7,9], which can hardly be explained by the hydration of eclogite in previous models based on water-enhanced electrical conductivity of emphacite[14] unless an unrealistically high water capacity is assumed (0.7 wt.% $H_2O$).

## Methods
### Geological background and sample preparation
The Dabie–Sulu orogenic belt was formed by the Triassic subduction of the Yangtze craton beneath the Sino-Korean lithospheric plate[58,68–70]. The Dabie–Sulu orogen is separated into two terranes by approximately 500 km of left-lateral strike-slip displacement along the Tan–Lu fault (Supplementary Fig. 1). The Dabie terrane in the west is the major segment bounded by the Tan–Lu fault and separated into a series of continuous zones by several large-scale EW-trending faults. The Sulu terrane in the east is segmented into a number of blocks by several NE-SW trending faults subparallel to the Tan–Lu fault. The Dabie-Sulu HP-UHP belts are unconformably overlain by Jurassic clastic sedimentary rocks and Cretaceous volcaniclastic deposits, and are intruded by post-collisional Mesozoic granites[59]. The continental crust subducted to the mantle at 80–200 km depths as indicated by the coesite and diamond samples in the Dabie–Sulu UHP metamorphic belts, followed by large-scale UHP metamorphism and subsequent exhumation to the shallow crust demonstrated by the coesite and

microdiamond inclusions within zircon from gneisses[58,69,70]. The eclogites in the Dabie–Sulu UHP terrane can be divided into three types based on their occurrence and host rocks. They are, Type I as enclaves or layers in granitic orthogneisses, Type II as enclaves in or interlayered with marbles, and Type III as enclaves and interlayers with ultramafic rocks[58,71]. Geochronological studies[59,69,72] indicated that the protolith ages of eclogites are 657–757 Ma for the Dabie belt and 747–788 Ma for Sulu belt, whereas metamorphic ages and exhumation ages are 225–235 Ma and ~230 Ma, respectively (Supplementary Table 3).

Four eclogite rock samples were used in this study: two (DB11 and DB13) collected from outcrops in Bixiling area, which is in the Central Dabie UHPM belt, and two (SL9 and SL12) collected from outcrops in Zhucheng and Rongcheng regions, both of which are in the northern part of the Sulu UHPM belt (Supplementary Fig. 1). These four samples were also used in previous study[71]. The mineral assemblage and major elements were reanalyzed in this study. All of the samples were fresh, fine-grained, and free of strong retrograde alteration. The concentrations of amphibole, epidote, biotite, aegirine, which are caused by garnet -> biotite + epidote, and omphacite -> amphibole + albite, are still very low (<4%) (Supplementary Table 3). Disks with a 6.0-mm diameter and 1.5-mm thickness were cored from the raw rocks, polished with sandpaper and 1-μm diamond powder, and dried in a vacuum oven at 473 K for 24 h before assembly.

### Mineral assemblage analysis
The modal mineral abundances in each sample were determined by point counting under a polarizing microscope on a series of thin sections. The primary mineralogy of these samples is garnet (40–60 vol%), omphacite (25–40 vol%), symplectite (10–20 vol%), and small amounts of accessory minerals (rutile, quartz, apatite, phengite, epidote, zoisite, and kyanite). Detailed petrographic features of these four samples are shown in Supplementary Fig. 2, and the peak metamorphic temperatures and pressures are listed in Supplementary Table 3.

### Major elements analysis
The chemical composition of each sample was determined by X-ray fluorescence spectrometry (Supplementary Table 4). The eclogitic rocks of Dabie and Sulu terranes showed a small variation in chemical composition. In a TAS (total alkali vs. silica) diagram (Supplementary Fig. 6), all of them fall in the category of picro-basalt and basalt. Geochemical studies indicated that the protoliths of Type I eclogites are basalts and that of Type III eclogites are continental gabbros[58,71]. Water in fresh gabbros and basalts store in the phencrysts (feldspar, pyroxene, etc.) and glasses, and the alteration minerals (amphibole, chlorite, etc.) formed by the water-rock interaction when exposed at the surface. The compositions of the garnet and omphacite phases were analyzed using an electron microprobe, which verified that both are chemically homogeneous. The concentrations of the major components in garnet and omphacite are summarized in Supplementary Table 5.

### Infrared spectroscopy measurements
The water contents in the samples before and after the thermal property measurements were analyzed by Fourier-transformation infrared (FT-IR) spectroscopy using a Jasco FTIR-6200 Equipper with an IRT-7000 infrared microscope. The aperture size was 50 × 50 μm, which is sufficiently large to incorporate dozens of grains. More than five spots were measured on each constituent mineral using unpolarized light. Measurements were performed through optically clean, inclusion- and crack-free areas at 1 atm and 300 K with a continuous dry $N_2$ gas flow.

The water content in each mineral was calculated using the Beer–Lambert law $C_{H_2O} = \Delta/(I \times t \times \gamma)$[73], where $C_{H_2O}$ is the content of hydrogen species (wt. ppm $H_2O$), $\Delta$ is the integrated area of the -OH absorption bands ($cm^{-2}$), $I$ is the integral specific absorption coefficient

(ppm$^{-1}$ cm$^{-2}$), $t$ is the thickness (cm, double-side polished to 0.12 cm prior to FT-IR analysis), and $\gamma$ is the orientation factor discussed by Paterson[74]. The -OH absorption bands were integrated between 2800 and 3800 cm$^{-1}$ for garnet and between 3000 and 3800 cm$^{-1}$ for omphacite to obtain the Δ values. Integral specific absorption coefficients of 1.39 and 7.09 ppm$^{-1}$ cm$^{-2}$ were used for garnet and omphacite, respectively[73]. An orientation factor $\gamma$ of 1 was applied for garnet and 1/3 for omphacite[74]. The bulk water content in eclogite (Supplementary Table 6) was estimated from the volume fraction and water content of each constituent mineral. The samples showed nearly identical FT-IR spectra before and after the thermal property measurements (Supplementary Fig. 4), thus indicating no water loss during the experiments. However, the general uncertainty of FTIR analysis is 30–50% caused by a number of reasons including unpolarized infrared on unoriented minerals, baseline correction, sample thickness uncertainty, and so on. The variation of water content in this study is indeed within the analytical uncertainty.

## Thermal property measurements

**Theory of experimental thermal property measurements.** Heat transport in solids could occur by the vibration of atoms in the lattices and through the grain-boundaries. The vibrations of atoms about their equilibrium positions (crystal lattice) are strongly coupled with neighboring atoms. The vibrational energy is thus dissipated through excitation of adjacent atoms. A realistic model accounting for the quantization of the vibrations is referred to as phonons. Heat is thus transferred through collision of phonons with each other. With either increasing pressure or decreasing temperature, the vibration frequency will increase, leading to the increase of phonon collision frequency. As a result, the thermal conductivity increases.

In the Debye's model, the contribution of phonon scattering to thermal conductivity ($\kappa_{lat}$) is approximately expressed by the relation[29,75]:

$$\kappa_{lat} = \frac{1}{3} C_V \nu \rho l \tag{3}$$

where $C_V$ is the volume specific heat capacity, $\nu$ is average phonon velocity (approximately equal to the sound velocity of solids), $\rho$ is the density, and $l$ is the mean free path. The mean free path is an important parameter to evaluate the ability of heat transfer. For example, if the density, specific heat capacity, and sound velocity reported in garnet[30,76] and omphacite[20,77] are considered in Eq. (3), the mean free path ($l$) is estimated to be about 5.5 Å and 7.6 Å for garnet and omphacite, respectively, at 1 atm and 300 K. Even through the temperature rises to 800 K, the $l$ value of omphacite (-3.5 Å) is still greater than that of garnet (-2.6 Å). A larger mean free path implies a higher $\kappa$ or $D$, which may reflect the influence of crystal structure on heat transport properties.

In addition to transport by phonons, heat could be transport by thermal radiation through a transparent and little-absorbing medium as electromagnetic waves (photons). The radiative heat transfer ($\kappa_{rad}$) depends on temperature, frequency of the electromagnetic waves, and three different length parameters (i.e., the size of the particulates involved in physical scattering, the distance over which about half of the photon flux is absorbed, and the distance over which temperature changes significantly)[78]. According to the blackbody radiation theory, $\kappa_{rad}$ increases with the cube of temperature[75,78]. At ambient pressure, $\kappa_{rad}$ of ferrous minerals approximately follow the equation[75]:

$$\kappa_{rad} = 8.5T^3/10^{11}[W/(mK)] \tag{4}$$

Therefore, the total thermal conductivity (effective thermal conductivity, $\kappa_{eff}$) of a solid can be written as

$$\kappa_{eff} = \kappa_{lat} + \kappa_{rad} \tag{5}$$

Generally, because of negative and positive temperature dependences of $\kappa_{lat}$ and $\kappa_{rad}$, respectively, $\kappa_{eff}$ is dominated by $\kappa_{lat}$ at relatively low temperature conditions (<1000 K, corresponding to lithospheric temperature conditions), and by $\kappa_{rad}$ at higher temperatures.

**High-pressure experiments accompanied with thermal property measurements.** The $D$ and $\kappa$ of the samples were measured on a DIA-type multi-anvil apparatus using a cell assembly design (Supplementary Fig. 7) following our previous studies[14,28,79]. Samples were isolated from a graphite heater using an alumina sleeve, which also acted as a thermal insulator to effectively restrict lateral heat flow. Three identically double-side polished disks were piled at the center of a pyrophyllite cubic pressure medium. A K-type (NiCr−NiAl) thermocouple and an impulse heater (Ni or Mo) were set on each interface between two disks. Nickel rods in contact with the samples were used as heat sinks to ensure a constant temperature boundary condition.

In-situ measurements of $D$ and $\kappa$ were performed at pressures up to 3 GPa and temperatures up to 873 K using the transient plane-source method[30]. The assembly was compressed to the desired pressure at room temperature, followed by heating to the target temperature over approximately 60 min. The assembly was then cooled at a rate of 10 °C/min, during which the $D$ and $\kappa$ were measured every 50 or 100 K. More than three repeated measurements at each temperature were performed to check for data reproducibility. The pressure was calibrated using the phase transition of Bi (2.54 GPa at room temperature) and melting of halide (high temperature). The uncertainties of the pressure estimation and temperature measurement with a K-type thermocouple are approximately 0.2 GPa and 1 K, respectively. To avoid any possible dehydration of garnet and omphacite and melting of eclogite, the experiments were performed at temperatures below 873 K.

**Data acquisition.** The thermal disturbance caused by impulse heating was monitored by the thermocouple under each pressure and temperature condition. The temperature variation (Δ$T$) at the thermocouple position can be expressed as a function of time ($t$)[30,80]:

$$\Delta T = A \sum_{n=1}^{\infty} \frac{1}{n^2} \sin\frac{n\pi}{3} \sin\frac{n\pi d}{h} \exp{-n^2Bt}[\exp(n^2B\tau) - 1] : t > \tau \tag{6}$$

where $\tau$ is the duration of impulse heating (s), $d$ is the distance between the impulse heater and thermocouple (m), and $h$ is the total height of the three sample disks (m). The parameters $A$ and $B$ are defined as:

$$A = \frac{2Qh}{\pi^2\kappa S}, \quad B = \frac{\pi^2 D}{h^2} \tag{7}$$

where $Q$ is the power of the impulse heating (W), $S$ is the area of the impulse heater (m$^2$), $\kappa$ is the thermal conductivity (Wm$^{-1}$ K$^{-1}$), and $D$ is the thermal diffusivity (mm$^2$ s$^{-1}$).

The temperature-time curve (Δ$T$-$t$) was obtained in the experiments, whereas the parameters $A$ and $B$ were determined through the least-square fitting of the Δ$T$-$t$ curves in Eq. (6) by sweeping $n$ up to 15, whose reliability has been verified by finite element simulations[81]. Once $A$ and $B$ are obtained, $D$ and $\kappa$ can be calculated from Eq. (7).

The errors of the experimental results of $D$ and $\kappa$ mainly come from the uncertainty of the sample thickness $h$, heating area $S$, pulse heating power $Q$, temperature, and the fitting errors of $A$ and $B$. The sample geometry during compression and heating was corrected based on the equation of state of eclogite[82] with the assumption of

isotropic contraction of the eclogite sample. The change of impulse heater area was calculated using the method proposed by Wang et al.[20]. The temperature disturbance across the sample associated with pulse heating was ~3 K with a 10-W pulse power. The effect of temperature heterogeneity on the measured results was thus negligible. Our previous work[79] also demonstrated a negligible influence of sample thickness and heating history (heating or cooling cycles) on the experimental results using the transient plane-source method. The total experimental uncertainties of $D$ and $\kappa$ originating from the abovementioned factors are therefore expected to be less than 5% (Supplementary Table 1).

As discussed above, the experimentally determined $\kappa$ (and D) based on the contact method contains the contributions of both $\kappa_{lat}$ and $\kappa_{rad}$. Practically, it is difficult to distinguish them in experiments[29,75,83]. However, because our experiments were performed under relatively low temperature conditions, the contribution of $\kappa_{rad}$ is limited. Based on Eq. (4), $\kappa_{rad}$ is less than 3% of $\kappa_{eff}$. Therefore, the contribution of $\kappa_{rad}$ has negligible effect on $\kappa_{lat}$ determined in this study.

### Simulation of geotherm profiles in the lithosphere

The crust and lithosphere are divided into a finite number of thin layers of thickness $\Delta z$ for each layer considering steady-state conductive heat transfer in a medium with arbitrary heat production. The heat generation ($A$) and thermal conductivity ($\kappa$) are assumed to be constant in each layer. The temperature profile within each layer is computed by a one-dimensional model using a boot-strapping method[46,47]:

$$T(z) = T_T + (Q_T/\kappa) \times z - (A/2\kappa) \times z^2 \qquad (8)$$

where $T_T$ and $Q_T$ are the temperature and heat flow at the top of the layer, respectively, and $z$ is the depth within the layer.

The temperature and heat flow at the bottom of the layer ($T_B$ and $Q_B$, respectively) can be calculated from $T_T$ and $Q_T$:

$$T_B = T_T + (Q_T/\kappa) \times \Delta z - (A/2\kappa) \times \Delta z^2 \qquad (9)$$

$$Q_B = Q_T - A \times \Delta z \qquad (10)$$

in which $T_B$ and $Q_B$ are set as the $T_T$ and $Q_T$ values for the underlaying layer.

By applying Eqs. (8)–(10), the geotherm profiles are calculated from the surface downward using a depth increment of 10 m in each step ($\Delta z = 10$ m). Following this approach, the crustal structures, composition, and heat production in the Sulu UHPM belt[50,51] and Himalaya-Tibet orogenic belt[48,49,52,84,85] are shown in Supplementary Fig. 5, and the temperature profiles are given in Fig. 3. The $\kappa$ of ultrahigh-pressure eclogite, granite, gneiss, amphibolite, and granulite are taken from this study and references[28,41,42]. Two alternative heat productions were assumed in the lithospheric mantle[51]: a depleted mantle with $A = 0.02\ \mu W\ m^{-3}$; and an enriched mantle with $A = 0.18\ \mu W\ m^{-3}$. Note that the surface heat flow $Q_T$ exerts the most significant influence on the corrected and uncorrected geotherms in Eq. (5), and consequently on the lithosphere thickness. $Q_T$ is 65 mW m$^{-2}$ for the Sulu belt and 75 mW m$^{-2}$ for the Tibetan Plateau based on the database of surface heat flow[44,45]. The mantle adiabat was set to a potential temperature of 1573 K with a gradient of 0.3 K/km, and the surface temperature was fixed at 283 K.

### Data availability

The data that support the findings of this study are available from the corresponding authors upon request.

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

## Acknowledgements

We appreciate the help of Dr. Hongfeng Tang for thin section identification, Dr. Xiaozhi Yang for FTIR measurements and Drs. Jia Liu, Xiaoyan Gu, Yantao Hao, and Wancai Li for helpful discussions. This study was financially supported by the Basic Science Center Program for Multiphase Media Evolution in Hypergravity of the National Natural Science Foundation of China (51988101), the National Natural Science Foundation of China (41973056, 41773056) to B.H.Z., the Fundamental Research Funds for the Central Universities (K20210168) to Q.K.X., and Key Research Program of Frontier Sciences of CAS (ZDBS-LY-DQC015) to B.H.Z.

## Author contributions

B.H.Z. conceived the idea and designed the experiments. B.H.Z. and H.Z.F. wrote the paper. J.H.G. performed the experiments. B.H.Z. and J.H.G. analyzed the data. L.S.Z. and Q.K.X. contributed to the data interpretation. All authors participated in the discussion and agreed on the content.

## Competing interests

The authors declare no competing interests.

## Additional information

**Supplementary information** The online version contains

supplementary material available at https://doi.org/10.1038/s41467-022-32484-w.

