## [Peer Review File · Nature Communications]

REVIEWER COMMENTS

Reviewer #1 (Remarks to the Author):

Review of „Crustal melting in orogenic belts revealed by eclogite thermal properties “ from Zhang et al.

The authors provided a study on the determination of thermal conductivity and diffusivity of natural eclogite, the dominant component of the thickened continental crust, as a function of pressure, temperature, composition, and water content, and simulated the thermal structure of the crust and lithosphere beneath orogenic belts. The multi-methodological study is generally well written. It opens with a short, but clear introduction. The overall structure is understandable. The figures and tables are of good quality, although some improvements are possible.

My recommendation is acceptance with minor revision.

Key results

Show experimental evidence that crustal partial melting occurs in a very shallow depth of around 30 km, which provides a thermal explanation for the low seismic velocity and high electrical conductivity at corresponding depths beneath thickened continents.

Data and methods

The authors are using a DIA-type multi-anvil apparatus using a cell assembly design. I would like to point out the discussion in Förster et al. (2020), who stated: “contact-based methods are thought to fall short of providing accurate measurements of phonon transport, causing systematic and opposing errors, due to both thermal resistance at contact interfaces and the presence of unwanted direct radiative transfer (Hofmeister, 1999, 2007)” I would like to invite the authors to reflect on the question of unwanted direct radiative transfer as discussed by Hofmeister, 1999, 2007 and its potential impact on their results.

Regarding the used heat flow database, I recommend to the authors to switch to the official database of the international heat flow commission (www.ihfc-iugg.org) or to reviewed and published data, like the one from Lucazeau, 2019.

General remarks:

Figures 1+2: Smaller symbols and error bars attached to the measurement points would improve the information content of the figures. Give the R^2 value for the fit of trend lines and data.

References

- Förster et al., 2020: <https://doi.org/10.1016/j.geothermics.2020.101937>
- Hofmeister, A.M., 1999. Mantle values of thermal conductivity and the geotherm from phonon lifetimes. *Science* 283, 1699–1705.
- Hofmeister, A.M., 2007. Pressure dependence of thermal transport properties. *Proc. Natl. Acad. Sci.* <https://doi.org/10.1073/pnas.0610734104>.
- Lucazeau, 2019: <https://doi.org/10.1029/2019GC008389>

Reviewer #2 (Remarks to the Author):

Review of # NCOMMS – 21 – 51133 – T – Crustal melting in orogenic belts revealed by eclogite thermal properties.

The authors' aims for writing this manuscript which are: to know whether the temperature of thickened crust is sufficiently high to induce partial melting, to determine the effects of pressure, modal mineralogy and water on the thermal diffusivity and thermal conductivity of eclogite, are valid motivations to write the manuscript. Achievement of these aims will contribute to the geoscientific knowledge of the Dabie-Sulu ultrahigh-pressure metamorphic belt in eastern China. The manuscript is technically valid. However, the discussion of the results did not vividly tackle the aims of the research. The authors did not thoroughly explain the effects of geothermal properties of eclogite and its implication to partial melting and continental behaviors. The discussion failed to connect in detail the relationship between pressure, temperature, modal and mineral composition and how they control thermal diffusivity and conductivity values of eclogite which in turn reflect crustal conditions.

There are problems with the organization of sections of the paper, lack of sufficient explanation in many places and with interpretation of findings in the discussion and conclusions. The authors based on the high lithospheric temperature to conclude on the partial melting of the study area. The discussions and conclusions are not informative enough.

My edits includes examples of some of the major issues and questions to help the authors address them.

Firstly the abstract contained references which is usually is not allowed. Abstract are not supposed to have references. The abstract concluded with other people's work instead of the authors own findings (line 38 to 40).

Also arrangement of the section headings is irregular and not numbered, they did not follow the normal format for paper presentation. Results and references were presented before the methods used. Geological map to show location of the study area and samples is omitted and there is no background information on the geology, geodynamics and tectonic setting of the Dabie-Sulu ultrahigh-pressure metamorphic belt in eastern China.

On the samples used for the experimental work, the kind and the nature of the sample were not disclosed. Whether the samples are grab or Diamond drill holes samples there is no idea. Two samples per location is inadequate for such experimental analysis and to extend the findings of analysis performed on four samples (local scale) to global scale is statistically inappropriate. This sample size may not be a true representation of the crustal and lithospheric conditions of the study area.

The topic indicates that, study was done on more than one belts but in actual fact it was done on only one belt (Dabie-Sulu ultrahigh-pressure metamorphic (UHPM) belt) which is separated by fault into eastern and western section. I suggest modification of the topic to reflect a belt instead of belts.

Line specific comments:

Pg. 3 line 64: This sounds like what the authors aimed out to determine, however they did not stress on how relevant this is to the geoscientific world. No references to previous works which are similar to their research work.

Pg. 4 line 70: From the statement this is one belt and not belts. Hence the topic should be modified to reflect this statement.

Pg. 4 line 73: This assertion is wrong. Eclogite properties vary per tectonic settings and cannot relate results on eclogite from Dabie and Sulu to all continental crusts worldwide.

Pg. 4 line 74: This is statement pertaining to this work? If it is, then it has come too early. It should not be in the introduction section.

Pg. 4 line 76: Discussion heading should come before line 76.

Pg. 4 line 78: What are these given pressures?

Pg. 4 line 81: How low is low temperature and how high is high temperature it will be appropriate to give values of low and high temperatures. Please explain why D and K increase in pressure despite temperature varies.

Pg. 5 line 101: This contradicts earlier statement in line 81. Not too clear what you want to say.

Pg. 5 line 102: Why do you attribute this to lattice conduction? Any explanation?

Pg. 5 line 114: Can you please let your readers know why garnet being more than omphacite in the eclogite has lower D and K. Explanation can be in terms of the mineralogical, petrophysical characteristics and crystal density of garnet.

Pg. 5 line 115: Quite a number of work have been done to explain the effects of water on D and K. To have a good discussion, mention these works and state how similar or different your results are to theirs. Where from the water? Is it part of the crystal structure of omphacite or part of the crustal environment? It is still not clear what the protolith of eclogite in the study area is. Knowledge of the protolith will help answer for the source of water in eclogite considering the P-T path transformation during metamorphism. For instance, the transformation of gabbroic rock of epidote and glaucophane composition from granulite to eclogite facies will release water. Also transformation of protolith (crustal rocks) from blueschist to eclogite facies that is associated with lawsonite or clinozoite releases the most of water during subduction. So it means that the presence of these mineral assemblages (hydrous minerals) in the protolith rock couple with your experimental analysis will provide you with strong argument to have a good discussion on the effect of water on the thermal properties of eclogite.

Pg. 5 line 118: You are discussing the effect of water on eclogite, and all of a sudden you relate your result to previous works done on olivine. You should relate your observations to previous works done on eclogite first, then establish any relationship between eclogite and olivine that will prove the effect of water on eclogite. What do you mean by negative water content?

Pg. 5 line 129: I found this to be untrue. Is this statement pertaining only to the study areas or it is a general? There are researches on partial melting regarding these tectonic activities, be it subduction, collision or exhumation.

Pg. 7 line 136: This statement needs to be referenced.

Pg. 7 line 148-149: Give the depths at which the geotherm occurred for both Tibetan and Sulu separately first before giving a range of depth. It was also observed from Fig. 3 the geotherm coincides with the Moho temperature. Any explanation? Please explain this in detail.

Pg. 7 line 141: Was there any radiogenic work in this paper? If no then reference the statement in line 141 to 144 properly.

Pg. 7 line 149 to 156: The discussion here is porous. Mention the temperatures of all the solidus (solidus temperature of crustal rocks, solidus temperature of wet granite, solidus temperature of amphiboles and peridotite). Then mention the corresponding temperatures from your results at the various depth profiles, (43, 55 and 103km) that compare well with the solidus. Give reasons, a part from the geotherms to support why partial melting will start both in the lower crust and at the deeper part of the lithosphere. Example, the presence of dehydration reaction products in the rocks. The essence of 50wt. ppm H₂O is also an evidence which can be argued out well to support your findings. Dehydration of melting of amphibolite generate a lot of water. Geothermal structures alone cannot be

used to prove partial melting. Other evidences must be factored in your discussion. Example: the micro textures (exsolution texture) as seen in the thin sections, presence of index minerals, coesite or minor diamond inclusions, and metamorphic mineral assemblages should all be used to support your experimental results on partial melting. Knit them together to make a good discussion.

Pg. 8 line 159 to 187: The flow of argument in the discussion is distorted. This is confusing because one cannot tell whether the authors were discussing their own results or other people's results. I advise the authors learn to discuss their own outcomes first, then support it by citing previous works. Line 159, the authors were of the view that, lowermost crust should retain high temperature. This should be suggested by low thermal diffusivity from their experimental result first before citing 27,40,41, and they went ahead to confirm this by a field data from Mesozoic Sulu orogeny which was not their own but by 34,36. Whatever this field data is, was not mentioned. Please specify the field data.

Move the statement from Experimental in line 160 up to line 163 to page 7, line 155.

Pg. 8 line 167 to 174: Which part of your work showed viscosity? Let us know. Other than that all you said are just speculations.

Pg. 8 line 177: Was there any seismic reflection or MT work? There was no mention of it in your methodology neither was there a presentation of results. I find it difficult to understand why you will be discussing other people's work instead of yours. You were of the impression that, lot of works were not done on the study area regarding your aims. But here you are talking a lot about previous works in your discussion.

Pg. 9 line 181: The work did not show any electrical conductivity done and the directions of flow, and again here are the authors discussing electrical conductivity of other people's works.

Pg. 9 line 187: It is not clear if this is your conclusion. The conclusion is

Figures

The authors have very good data and valid results from their experiments, this is evidenced from all the figures they produced. Unfortunately they could not interpret their data and results deeply enough.

From Fig. 1 and 2, the maximum temperature to determine the thermal properties of eclogite is 823K. Is there any reason why it could not be extended to the geotherm of 940K and beyond? Kindly let your readers have this information if you can.

Fig. 3: This figure must be well explained to resolve the issue raised on page 7, line 148 and 149. Here it can be seen that, in Tibetan the geotherm of 940K occurred within the depth of 25-30km, a little bit shallower than the depth of 30-35km in Sulu. This is because, the crustal thickness and the Moho discontinuity temperature of Tibetan 1340K is higher than that of Sulu. Also add that, the geotherm coincides with the Moho temperature of Sulu.

Extended Data Fig. 4

The authors must include the following information to the FT-IR spectroscopy analysis on the five spots measured to the extended data Fig.4 on page 28. Create a table and show in the columns the values for all the parameters in the Beer-Lambert law for each sample.

- i. The content of hydrogen species
- ii. The integrated area of the -OH absorption bands
- iii. The integral specific absorption coefficient

The OH content of omphacite helps determine the thermal properties of eclogite, hence the authors should elaborate more on the OH content of omphacite from their FT-IR spectroscopy in their discussion. Any dissolution of H₂ into OH must be stated. What is the temperature and pressure of the spectra analysis?

Samples DB 11 and DB 13 show weak peaks for any quantitative evaluation. Please any reasons for this?

Tables

Extended Data Table 2

How is the modeled data correlating with the experimental data? What is T (temperature) values in this Table? Is it measured in K or °C?

Extended Data Table 3

Apart from the samples IDs which belongs to the current work, and guess of geochronology by previous workers, the Table did not indicate which part belongs to the current research and which belongs to the previous work 60. It is also not clear whether the same samples were used by previous workers.

Extended Data Table 4

No geochemical deductions were made from the major element compositions in this Table.

Classification plots can be drawn from this data.

Extended Data Table 5

The water content after run on garnet for samples SL9, DB11 and DB13 have increased and SL12 has decreased. What could be causing this change? Why two samples from the same location have this disparity?

Reviewer #3 (Remarks to the Author):

This paper presents new experimental data on the thermal properties of natural eclogite, that are used to refine the thermal structure of orogenic crust in continental collision environments, and to discuss implications on partial melting (anatexis), which plays an important role on the rheology of the orogenic crust.

The new data are derived from rigorous and comprehensive experimental work, that consider the role of P - T , rock composition and H₂O content, and are placed within the framework of other data on eclogites. Furthermore, they provide the basis for numerical modeling to compare the thermal structure and anatectic potential of two orogenic systems with contrasting crustal thicknesses, the Dabie Shan/Sulu and the Tibetan plateau.

The results improve our understanding of thermal structure in orogenic belts, and inform on why orogenic continental crust undergoes partial melting. I focus the rest of my review in some aspects of concern.

Refers to line numbers

- 1- The manuscript gives the impression that partial melting of orogenic crust is still a controversial issue and that whether the continental crust can reach sufficiently high temperatures to produce partial melts, is questionable (ex. see #29–30, 54–55, 132–133). I agree that there is still a lot to be learned on this topic, but there are numerous petrological studies from crustal anatectic rocks from continental collision environments, and the authors mention the existence of independent evidence for melting in #130–131. In this context, this contribution is important not because it makes a case that anatexis can happen (something that we know anyways) but because it provides insight on why orogenic crust can partially melt.
- 2- There is some ambiguity in the use of certain terms and in some statements made. For instance:
 - (a) Continental collision results in crustal thickening and subduction of continental crust, but these two are not necessarily the same. This paper focuses on subduction, but crustal thickening and subduction are used interchangeably and this leads to some confusion.
 - (b) Eclogite is a major component of subduction zones (as stated in #49) and of deep levels of thickened continental crust, but it is not the dominant component of the thickened continental crust (as stated in #32)
- 3- Some critical information for the assessment of the manuscript is missing: this pertains to the age of high- P metamorphism and exhumation of the Dabie /Sulu belt, where the investigated samples come from. This is particularly important because the results are used to model and compare with the thermal structure of the Tibet, and because it is stated that the lithosphere in Dabie/Sulu there is thin (see #135–136). I guess this is a feature that developed later in the evolution of the belt?

Other

#34–37 30 km depth is close to the Moho discontinuity in ‘normal’, un-thickened continental crust; however, the calculations were done for thick, orogenic crust?

#47–48 *‘Early studies attributed these features to the hydration of eclogite....’* The reference in support to this (¹⁵ Liu et al., 2019) is a rather modern one...

#130–131 Felsic rocks also occur in these settings; they experience anatexis at lower pressures than mafic rocks, and their melting would also influence the rheology of the crust....

139–140 *‘..., the Moho temperature of the Sulu belt (~940 K) (Fig. 3a) is considerably lower than that of the Tibetan Plateau (~1340 K) (Fig. 3b), which implies a hotter Tibetan lithosphere.’* But also the Tibetan plateau Moho is much deeper than that under Sulu.... In fact for the same depth, Sulu lithosphere is hotter (see Fig. 3)

#148–156 This part is difficult to read without information about the evolution of the Dabie /Sulu belt, and on when was the thin lithosphere achieved (see also main point 3)

#159–160 *‘the lowermost crust should be able to retain a protracted high temperature, as suggested by its low thermal diffusivity’* I don’t understand this statement; this doesn’t also depend upon the rate of tectonic processes?

Online content

#321 How is a ‘strong’ retrograde alteration is defined?

331 It would be very useful to add ages of peak metamorphism and exhumation in extended Table 3.

Feb. 26, 2022

A. Indares

Response to reviewers' comments

Reviewer #1 (Remarks to the Author):

The authors provided a study on the determination of thermal conductivity and diffusivity of natural eclogite, the dominant component of the thickened continental crust, as a function of pressure, temperature, composition, and water content, and simulated the thermal structure of the crust and lithosphere beneath orogenic belts. The multi-methodological study is generally well written. It opens with a short, but clear introduction. The overall structure is understandable. The figures and tables are of good quality, although some improvements are possible.

My recommendation is acceptance with minor revision.

Key results

Show experimental evidence that crustal partial melting occurs in a very shallow depth of around 30 km, which provides a thermal explanation for the low seismic velocity and high electrical conductivity at corresponding depths beneath thickened continents.

We appreciate the reviewer for the positive evaluation of our work.

Data and methods

The authors are using a DIA-type multi-anvil apparatus using a cell assembly design. I would like to point out the discussion in Förster et al. (2020), who stated: "contact-based methods are thought to fall short of providing accurate measurements of phonon transport, causing systematic and opposing errors, due to both thermal resistance at contact interfaces and the presence of unwanted direct radiative transfer (Hofmeister, 1999, 2007)". I would like to invite the authors to reflect on the question of unwanted direct radiative transfer as discussed by Hofmeister, 1999, 2007 and its potential impact on their results.

1. We have added discussions about the effect of direct radiative transfer (\$\kappa_{\text{rad}}\$ ) on the accuracy of our experimental results (Line 112-115, 369-371).

As pointed out by Hofmeister (1999, 2007) and Förster et al. (2021), laboratory thermal property experiments contain both lattice thermal conductivity (\$\kappa_{\text{lat}}\$ ) and direct radiative transfer (\$\kappa_{\text{rad}}\$ ). However, the \$\kappa_{\text{rad}}\$ calculated from Eq. (4) given by Hofmeister (1999, 2007) is only about 3% of the \$\kappa_{\text{eff}}\$ measured in our experiments. Therefore, the effect of \$\kappa_{\text{rad}}\$ should be negligible in this study.

Regarding the used heat flow database, I recommend to the authors to switch to the official database of the international heat flow commission (www.ihfc-iugg.org) or to reviewed and published data, like the one from Lucazeau, 2019.

2. We have accordingly corrected it in the revised manuscript (Line 153, 396).

General remarks:

Figures 1+2: Smaller symbols and error bars attached to the measurement points would improve the information content of the figures. Give the R^2 value for the fit of trend lines and data.

3. The error bars are smaller than the symbols, i.e., less than 1.5% (Supplementary Table 1). It is explained in the figure captions. We have given the R^2 value for the data fitting in Supplementary Table 2.

References

- Förster et al., 2020: <https://doi.org/10.1016/j.geothermics.2020.101937>
- Hofmeister, A.M., 1999. Mantle values of thermal conductivity and the geotherm from phonon lifetimes. *Science* 283, 1699–1705.
- Hofmeister, A.M., 2007. Pressure dependence of thermal transport properties. *Proc. Natl. Acad. Sci.* <https://doi.org/10.1073/pnas.0610734104>.
- Lucazeau, 2019: <https://doi.org/10.1029/2019GC008389>

4. We have cited these references and renumbered them in the revised manuscript.

Reviewer #2 (Comments to the Author):

The aims for writing this paper which are: to know whether the temperature of thickened crust is sufficiently high to induce partial melting, to determine the effects of pressure, modal mineralogy and water on the thermal diffusivity and thermal conductivity of eclogite, are valid motivations to write the manuscript. Achievement of these aims will contribute to the geoscientific knowledge of the Dabie-Sulu ultrahigh-pressure metamorphic belt in eastern China. The manuscript is technically valid. However, the discussion of the results did not vividly tackle the aims of the research. The authors did not thoroughly explain the effects of geothermal properties of eclogite and its implication to partial melting and continental behaviors. The discussion failed to connect in detail the relationship between pressure, temperature, modal and mineral composition and how they control thermal diffusivity and conductivity values of eclogite which in turn reflect crustal conditions. There are problems with the organization of sections of the paper, lack of sufficient explanation in many places and with interpretation of findings in the discussion and conclusions. The authors based on the high lithospheric temperature to conclude on the partial melting of the study area. The discussions and conclusions are not informative enough.

Thank you for your constructive views. Following the suggestions, we have made significant revision of the manuscript. The aims of this study are more clearly described (Line 27-28, 39, 57 and 178). The effect of P, T, water content, modal and mineral compositions on the thermal properties, and the connection between eclogite thermal properties, lithospheric temperature, and partial melting are more explained in greater details (Line 141-210). We also reorganized the sections as shown in Line 140-217.

My edits includes examples of some of the major issues and questions to help the authors address them.

Firstly the abstract contained references which is usually is not allowed. Abstract are not supposed to have references. The abstract concluded with other people's work instead of the authors own findings (line 38 to 40). Also arrangement of the section headings is irregular and not numbered, they did not follow the normal format for paper presentation. Results and references were presented before the methods used.

1. Following the suggestions, the abstract is modified and the references are deleted. The format of this manuscript was changed to follow the instructions of Nature Communications.

Geological map to show location of the study area and samples is omitted and there is no background information on the geology, geodynamics and tectonic setting of the Dabie-Sulu ultrahigh-pressure metamorphic belt in eastern China.

2. The geological background and geological map (Supplementary Fig. 1) are added in Line 220-238.

On the samples used for the experimental work, the kind and the nature of the sample were not disclosed. Whether the samples are grab or Diamond drill holes samples there is no idea.

3. Our natural eclogite samples were collected from outcrops in the Dabie-Sulu belt but not from Diamond drill holes. Related information about sample collection is provided. (Line 83-84, 239-241)

Two samples per location is inadequate for such experimental analysis and to extend the findings of analysis performed on four samples (local scale) to global scale is statistically inappropriate. This sample size may not be a true representation of the crustal and lithospheric conditions of the study area.

The topic indicates that, study was done on more than one belts but in actual fact it was done on only one belt (Dabie-Sulu ultrahigh-pressure metamorphic (UHPM) belt) which is separated by fault into eastern and western section. I suggest modification of the topic to reflect a belt instead of belts.

4. It's true that the eclogite samples are from limited locations. However, we should point out that the eclogite samples in this study show largely similar petrographic as geochemical features to those from the Dabie-Sulu as well as the other UHP orogenic belts worldwide, they are representative and could be used to address important questions of the processes responsible for the tectonic evolution of collisional orogenic belts worldwide.

In this revised manuscript, we slightly modified the manuscript to focus more on the Dabie-Sulu and the Himalaya-Tibet orogenic belts as two representative examples. (Line 157-202) Insights yield from this study could be applicable to the other collisional orogenic belts.

(1) In all continental collision orogenic belts worldwide, eclogite is mainly composed of garnet and omphacite. As summarized in Table S1, the chemical compositions of garnets and omphacites in the UHP eclogites from different orogenic belts (e.g., Dabie-Sulu, Bohemian, Western Norway, Papua New Guinea, Greenland and Kokchetav) display great similarities. Therefore, the eclogite samples from the Dabie-Sulu UHP belt can represent the eclogites in other orogenic belts.

(2) Because of the similarity of mineral compositions, the thermal conductivity and diffusivity of eclogite from various orogenic belts should also be comparable. Indeed, we found that the D and κ of eclogites are independent from garnet content (Figs. 2a and 2b) and weakly depend on omphacite content (Supplementary Fig. 3). In this regard, the thermal properties of eclogites should not vary from place to place.

(3) Although there are differences in the geometry (e.g. the angle and the rate) of subduction, it is widely accepted that eclogitization of subducted materials is a common HP to UHP metamorphic process. In this regard, the formation process of eclogite is similar for orogenic belts worldwide.

Table S1. Chemical compositions of garnets and omphacites in the UHP eclogites from different orogenic belts (in wt.%).

	D-S (SL9)		B (St11-7)		WN (419)		PNG (870921)		G (06-59)		K (K12A)	
	Grt	Omp	Grt	Omp	Grt	Omp	Grt	Omp	Grt	Omp	Grt	Omp
SiO ₂	38.72	52.85	38.77	55.32	38.92	51.63	39.86	56.24	39.35	53.11	39.19	53.78
TiO ₂	0.10	0.09	0.11	0.22	0.09	0.30	0.03	0.02	0.07	0.29	0.13	0.14
Al ₂ O ₃	19.68	7.84	22.28	13.44	22.21	6.31	21.69	12.95	22.40	11.96	22.12	5.09
Cr ₂ O ₃	0.02	0.01	0.00	0.07	0.08	0.00	n.a.	n.a.	0.00	0.01	0.00	0.04
FeO	22.83	6.11	16.32	2.83	21.49	6.55	23.27	4.90	18.82	5.25	14.08	5.53
MnO	0.36	0.03	0.42	0.06	0.61	0.00	0.64	0.03	0.26	0.01	3.95	0.24
MgO	6.24	10.59	6.24	7.98	5.98	12.50	8.87	7.10	8.21	8.88	6.02	12.36
CaO	12.38	15.64	14.51	14.15	11.30	19.78	6.43	11.50	11.31	16.10	13.84	20.48
Na ₂ O	0.06	6.91	n.d.	6.15	n.d.	1.95	n.d.	7.90	0.02	4.89	0.05	2.19
K ₂ O	n.a.	n.a.	n.d.	0.02	n.a.	n.a.	n.d.	n.d.	0.01	0.02	0.00	0.04
Total	100.39	100.07	99.99	100.24	100.66	99.01	100.79	100.82	100.45	100.53	99.38	99.89

Data sources: D-S: Dabie-Sulu (This study); B: Bohemian (Schmadicke et al., 1995); WN: Western Norway (Terry et al., 2000); PNG: Papua New Guinea (Baldwin et al., 2004); G: Greenland (Augland et al., 2010); K: Kokchetav (Zhang et al., 1997)

Based on above three reasons, we think that the thermal properties of eclogite from the Dabie-Sulu belt can be used to discuss other orogenic belts.

Line specific comments:

Pg. 3 line 64: This sounds like what the authors aimed out to determine, however they did not stress on how relevant this is to the geoscientific world. No references to previous works which are similar to their research work.

5. This sentence was rewritten as “However, the effects of pressure and modal mineralogy on the D and κ values of eclogite remain unknown, whereas such information is indispensable to infer the thermal structure of the continental

lithosphere and thus to interpret the generation of partial melting and metamorphic and geochemical geodynamics of collisional orogen.” (Line 66-69)

Pg. 4 line 70: From the statement this is one belt and not belts. Hence the topic should be modified to reflect this statement.

Pg. 4 line 73: This assertion is wrong. Eclogite properties vary per tectonic settings and cannot relate results on eclogite from Dabie and Sulu to all continental crusts worldwide.

6. As we explained above (point 4), eclogite samples from different belts have similar chemical compositions and modal mineralogy. Therefore, the K and D of eclogite determined in this study should be applicable for the other orogenic belts worldwide. In the discussion section, we focus on the Dabie-Sulu and Himalaya-Tibet belts as typical examples.

Pg. 4 line 74: This is statement pertaining to this work? If it is, then it has come too early. It should not be in the introduction section.

7. It is a summary of our conclusion. We agree this statement comes too early. So we deleted this sentence in the introduction.

Pg. 4 line 76: Discussion heading should come before line 76.

8. The heading “Results and discussion” was added before line 76. (Line 85)

Pg. 4 line78: What are these given pressures?

9. In line 78, “At a given pressure” was changed into “At 3 GPa”. (Line 88)

Pg. 4 line 81: How low is low temperature and how high is high temperature it will be appropriate to give values of low and high temperatures. Please explain why D and K increase in pressure despite temperature varies.

10. We mean at the same T conditions, D and K increases with increasing pressure. The corresponding sentence is reorganized as: “Figs. 1c and 1d clearly show that both D and κ increases linearly with increasing pressure at given temperature conditions (300 and 823 K), whereas the pressure derivative ($\partial D/\partial P$ and $\partial \kappa/\partial P$) is steeper at 300 K than 823 K.” (Line 91-93)

Pg. 5 line 101: This contradicts earlier statement in line 81. Not too clear what you want to say.

11. In line 101 we are talking about the negative temperature dependence at given pressure conditions. In Line 81 we are talking about positive pressure dependence at

given temperature conditions. Basically, there is no contradiction. To avoid any possible misunderstanding, we have revised the two sentences:

“The continuous decreasing of D and κ with increasing temperature up to 873 K in this study suggest that the thermal properties of eclogite is dominated by κ_{lat} .” (Line 110-112)

“Figs. 1c and 1d clearly show that both D and κ increases linearly with increasing pressure at given temperature conditions (300 and 823 K), whereas the pressure derivative ($\partial D/\partial P$ and $\partial \kappa/\partial P$) is steeper at 300 K than 823 K.” (Line 91-93)

Pg. 5 line 102: Why do you attribute this to lattice conduction? Any explanation?

12. We have revised the sentence to make it easier to read and understand (Line 109-115).

We attribute it to lattice conduction based on the following two reasons:

(1) Radiative thermal conductivity (κ_{rad}) is proportional to the cube of temperature, whereas lattice thermal conductivity (κ_{lat}) is inversely proportional to temperature (Hofmeister, 1999, 2007). Our results show decrease of κ with increasing temperature (Fig. 1) and therefore it is expected to be controlled by lattice conduction.

(2) The calculation from Eq. 4 (Hofmeister, 2007) suggests that the contribution of κ_{rad} to the experimentally determined κ_{lat} is negligible at the temperature range in this study.

Pg. 5 line 114: Can you please let your readers know why garnet being more than omphacite in the eclogite has lower D and K . Explanation can be in terms of the mineralogical, petrophysical characteristics and crystal density of garnet.

13. In the viewpoint of mineral physics, garnet crystal is isotropic, while omphacite is anisotropic. Although the density and sound velocity in garnet are higher than those in omphacite, specific heat capacity of omphacite is much larger than that of garnet, which would result in a larger mean free path for omphacite. Larger mean free path means higher κ or D . (Line 126-129)

Pg. 5 line 115: Quite a number of work have been done to explain the effects of water on D and K . To have a good discussion, mention these works and state how similar or different your results are to theirs.

14. We agree there are quite a number of works performed on the water effect on D and K of other minerals and rocks. We mean that the water effect on D and K of eclogite is previously unknown. The related sentence is accordingly revised. (Line 130-131)

Pg. 5 line 115: Where from the water? Is it part of the crystal structure of omphacite or part of the crustal environment? It is still not clear what the protolith of eclogite in the study area is. Knowledge of the protolith will help answer for the source of water in eclogite considering the PT path transformation during metamorphism. For instance, the transformation of gabbroic rock of epidote and glaucophane composition from granulite to eclogite facies will release water. Also transformation of protolith (crustal rocks) from blueschist to eclogite facies that is associated with lawsonite or clinozoite releases the most of water during subduction. So it means that the presence of these mineral assemblages (hydrous minerals) in the protolith rock couple with your experimental analysis will provide you with strong argument to have a good discussion on the effect of water on the thermal properties of eclogite.

15. In this study, the protoliths for eclogites are from basalts and gabbros (Tang et al., 2007; Zhang et al., 2009). (Line 71, 260-261)

Water in fresh gabbros and basalts is stored in the phenocrysts (feldspar, pyroxene, etc.) and glasses, and the alteration minerals (amphibole, chlorite, etc.) formed by the water-rock interaction when exposed at the surface. (Line 261-264)

Regardless of the protoliths (e.g., gabbros or basaltic rocks), eclogites are formed through high-pressure to ultrahigh-pressure metamorphism, and usually contain a limited amount of water in the constituted hydrous (e.g., phengite, lawsonite, epidote) and nominally anhydrous minerals (e.g., omphacite, garnet). Upon increasing of temperature during continental deep subduction, hydrous minerals breakdown, leaving omphacite and garnet as the main water reservoir (Zheng, 2012; Hermann et al., 2013; Zheng et al., 2016) (Line 71-75). Therefore, water in eclogites mainly exists as hydroxyl in the crystal structures of garnet and omphacite (Koch-Müller et al., 2004; Katayama et al., 2006; Liu et al., 2019; Zhang et al., 2019). (Line 135-136)

Pg. 5 line 118: You are discussing the effect of water on eclogite, and all of a sudden you relate your result to previous works done on olivine. You should relate your observations to previous works done on eclogite first, then establish any relationship between eclogite and olivine that will prove the effect of water on eclogite. What do you mean by negative water content?

16. Following the suggestion, the discussion about water effect on the thermophysical properties of eclogite was reorganized. (Line 130-139)

We do not mean “negative water content”, but “negative water-content dependence”. To avoid any misunderstanding, we revised this sentence to “In this

study, we demonstrate that the D and κ of eclogite decrease linearly with increasing water content (Figs. 2c, 2d)". (Line 131-132)

Pg. 5 line 129: I found this to be untrue. Is this statement pertaining only to the study areas or it is a general? There are researches on partial melting regarding these tectonic activities, be it subduction, collision or exhumation.

17. To avoid the possible misunderstanding, the related sentence is revised to "Numerous studies have provided petrological evidences for the occurrence of partial melting in natural UHP rocks during the continental collision (Zeng et al., 2009; Zheng et al., 2011; Sawyer et al., 2013; Xu et al., 2013; Kohn, 2014; Wang et al., 2014; Li et al., 2016). However, the temperature distribution beneath continental orogenic belts, which is essential for interpretation of the partial melt process during continental collision, is poorly understood." (Line 141-144)

Pg. 7 line 136: This statement needs to be referenced.

18. We have added references (Owens and Zandt, 1997; Li et al., 2020) in Line 161.

Pg. 7 line 148-149: Give the depths at which the geotherm occurred for both Tibetan and Sulu separately first before giving a range of depth. It was also observed from Fig. 3 the geotherm coincides with the Moho temperature. Any explanation? Please explain this in detail.

19. The calculated geotherm, Moho temperature, and the depth range of partial melting for the Sulu and the Himalaya-Tibet orogenic belts were discussed separately in the revised manuscript. (Line 157-172)

We do not mean "the geotherm coincides with the Moho temperature in Fig. 3." but "the Moho temperature is inferred from our calculated geotherm".

Pg. 7 line 141: Was there any radiogenic work in this paper? If no then reference the statement in line 141 to 144 properly.

20. Radiogenic work was from reference but not this study. We have added reference (Chi and Yan, 1998; He et al., 2009; Goes et al., 2020) in Line 163.

Pg. 7 line 149 to 156: The discussion here is porous. Mention the temperatures of all the solidus (solidus temperature of crustal rocks, solidus temperature of wet granite, solidus temperature of amphiboles and peridotite). Then mention the corresponding temperatures from your results at the various depth profiles, (43, 55 and 103km) that compare well with the solidus. Give reasons, a part from the geotherms to support why partial melting will start both in the lower crust and at the deeper part

of the lithosphere. Example, the presence of dehydration reaction products in the rocks. The presence of 50wt. ppm H₂O is also an evidence which can be argued out well to support your findings. Dehydration of melting of amphibolite generate a lot of water. Geothermal structures alone cannot be used to prove partial melting.

21. Following the suggestions, we have carefully reorganized the discussion section. We first describe in detail the calculated geotherm in this study, including the corresponding temperature at the various depths. Then we mention the solidus of hydrous minerals and rocks, and compare with the geotherm to show that partial melting can occur. Finally, we try to link between our geotherm profiles and the geological evolution process to explain more about when and why partial melting have occurred during subduction and exhumation of continental crust. (Line 140-217)

As to the hydrous peridotite, the related sentence is deleted in the revised manuscript because we have reorganized our discussion.

Pg. 7 line 149 to 156: Other evidences must be factored in your discussion. Example: the micro textures (exsolution texture) as seen in the thin sections, presence of index minerals, coesite or minor diamond inclusions, and metamorphic mineral assemblages should all be used to support your experimental results on partial melting. Knit them together to make a good discussion.

22. It is true that other evidences, including the micro textures (exsolution texture) in the thin sections and presence of index minerals (coesite or minor diamond inclusions), can be used to support the occurrence of partial melting. We have added and cited the corresponding references. (Line 187-192)

Pg. 8 line 159 to 187: The flow of argument in the discussion is distorted. This is confusing because one cannot tell whether the authors were discussing their own results or other people's results. I advise the authors learn to discuss their own outcomes first, then support it by citing previous works. Line 159, the authors were of the view that, lowermost crust should retain high temperature. This should be suggested by low thermal diffusivity from their experimental result first before citing 27,40,41, and they went ahead to confirm this by a field data from Mesozoic Sulu orogeny which was not their own but by 34,36. Whatever this field data is, was not mentioned. Please specify the field data.

23. Please see our above response (point 21).

Following the suggestions, we have reorganized the discussion section. We first describe in detail the calculated geotherm in this study, including the corresponding temperature at the various depths. Then we mention the solidus of hydrous minerals

and rocks, and compare with the geotherm to show that partial melting can occur. Finally, we try to link between our geotherm profiles and the geological evolution process to explain more about when and why partial melting have occurred during subduction and exhumation of continental crust. (Line 140-217)

Field data were mentioned and cited in corresponding places. (Line 187-192)

Move the statement from Experimental in line 160 up to line 163 to page 7, line 155.
24. We moved it.

Pg. 8 line 167 to 174: Which part of your work showed viscosity? Let us know. Other than that all you said are just speculations.

25. The related sentences are revised to make it more clear (Line 203-210).

We did not measure/calculate viscosity. We mean that, according to previous studies (Renner et al., 2000; Rosenberg and Handy, 2005), partial melt should significantly reduce the viscosity of rocks, and thus affect the geodynamic and geophysical processes. For example, partial melt is expected to weaken the subducted crust, leading to the decoupling of rocks from the subducting slab and resulting in the initiation of exhumation (Renner et al., 2000; Rosenberg and Handy, 2005; Labrousse et al., 2011). Partial melt may lubricate the edge of exhumated slices of UHP rocks, enhancing the transport of materials from deep regions to the surface (Labrousse et al., 2011; Zheng et al., 2011, 2016; Zheng, 2012; Hermann et al., 2013; Wang et al., 2014).” (Line 203-210)

Pg. 8 line 177: Was there any seismic reflection or MT work? There was no mention of it in your methodology neither was there a presentation of results. I find it difficult to understand why you will be discussing other people’s work instead of yours. You were of the impression that, lot of works were not done on the study area regarding your aims. But here you are talking a lot about previous works in your discussion.

26. Explanation of the previous MT data is one of our implication. Previous studies attribute the observations of seismic and MT studies to partial melting (Kind et al., 1996; Nelson et al., 1996; Wei et al., 2001; Unsworth et al., 2005; Nábělek et al., 2009; Yang, 2009; Xiao et al., 2009). However, previously it’s unknown why and how partial melting can occur in the orogenic continental crust. Our results provide thermal properties of eclogite, which is used to calculate the geotherm profile of the lithosphere, and our geotherm profile answers the question why partial melting can occur in the orogenic continental crust. The related argument is mentioned in Line 211-217.

Pg. 9 line 181: The work did not show any electrical conductivity done and the directions of flow, and again here are the authors discussing electrical conductivity of other people's works.

27. Please see our above response.

Pg. 9 line 187: It is not clear if this is your conclusion. The conclusion is

28. It is one of our conclusions. The last sentence was revised to "Therefore, the occurrence of partial melting in the lithosphere could explain the low seismic-velocity and high electrical conductivity observed in the lithosphere of Sulu (Yang, 2009; Xiao et al., 2009) and Himalaya-Tibet belts (Kind et al., 1996; Nelson et al., 1996; Wei et al., 2001; Nabelek et al., 2009; Unsworth et al., 2005; Bai et al., 2020), which can hardly be explained by the hydration of eclogite in previous models based on water-enhanced electrical conductivity of omphacite (Zhang et al., 2019) unless an unrealistically high water capacity is assumed (0.7 wt.% H₂O)." (Line 213-217)

Figures

The authors have very good data and valid results from their experiments, this is evidenced from all the figures they produced. Unfortunately they could not interpret their data and results deeply enough.

29. We thank the reviewer's constructive comments. Following the above suggestions, we have tried to interpret our data more deeply.

From Fig. 1 and 2, the maximum temperature to determine the thermal properties of eclogite is 823K. Is there any reason why it could not be extended to the geotherm of 940K and beyond? Kindly let your readers have this information if you can.

30. We have mentioned the reason in Line 339-340. It is because dehydration of garnet and omphacite and melting of eclogites could occur at higher temperature.

Fig. 3: This figure must be well explained to resolve the issue raised on page 7, line 148 and 149. Here it can be seen that, in Tibetan the geotherm of 940K occurred within the depth of 25-30km, a little bit shallower than the depth of 30-35km in Sulu. This is because, the crustal thickness and the Moho discontinuity temperature of Tibetan 1340K is higher than that of Sulu. Also add that, the geotherm coincides with the Moho temperature of Sulu.

31. We have modified Figure 3 following the suggestions. (Line 157-172)

Extended Data Fig. 4

The authors must include the following information to the FT-IR spectroscopy analysis on the five spots measured to the extended data Fig.4 on page 28. Create a table and show in the columns the values for all the parameters in the Beer-Lambert law for each sample.

- i. The content of hydrogen species
- ii. The integrated area of the –OH absorption bands
- iii. The integral specific absorption coefficient

The OH content of omphacite helps determine the thermal properties of eclogite, hence the authors should elaborate more on the OH content of omphacite from their FT-IR spectroscopy in their discussion. Any dissolution of H₂ into OH must be stated. What is the temperature and pressure of the spectra analysis?

32. A new table (Supplementary Table 6) is added to show all the parameters in the Beer-Lambert law for each sample.

The infrared measurement was performed at 1 atm and 300 K (Line 272-273). In eclogite samples, the speciation of H₂O is expected to be hydroxyl in the crystal structure of garnet and omphacite rather than other forms (Koch-Müller et al., 2004; Katayama et al., 2006; Liu et al., 2019; Zhang et al., 2019).

Supplementary Table 6 Infrared spectroscopy parameters and estimated water content (in wt. ppm) of garnets and omphacites using the Beer-Lambert law.

Parameter	DB11		DB13		SL9		SL12	
	Garnet	Omphacite	Garnet	Omphacite	Garnet	Omphacite	Garnet	Omphacite
$\Delta(\text{cm}^{-2})^{\S}$	1.97/2.02	3.77/4.09	0.43/0.80	1.49/1.75	4.59/4.66	10.26/9.71	3.52/2.60	6.32/4.26
$l(\text{ppm}^{-1} \text{ cm}^{-2})$	1.39	7.09	1.39	7.09	1.39	7.09	1.39	7.09
$t(\text{cm})^{\S}$	0.014/0.013	0.014/0.013	0.014/0.013	0.014/0.013	0.014/0.013	0.014/0.013	0.014/0.011	0.014/0.011
γ	1	1/3	1	1/3	1	1/3	1	1/3
$C_{\text{H}_2\text{O}}$ (wt. ppm) [§]	101/112	114/133	22/44	45/57	236/258	310/316	181/170	191/164
$C_{\text{H}_2\text{O}}$ (wt. ppm) ^{§§}	100/113		29/44		187/198		156/143	

The dissolution of H₂ is not expected in our samples. Because if molecular H₂ had presented, infrared absorptions at wavenumber of 4060-4100 cm⁻¹ should appear (Yang et al., 2016; Moine et al., 2020), which is not the case in this study (Fig. S1d). The related explanation is given in Line 53-54 of Supplementary Information.

Fig. S1 Unpolarized FT-IR spectra of (a) clinopyroxene (Yang et al., 2016), (b) garnet (Yang et al., 2016), (c) omphacite (Moine et al., 2020) containing dissolved molecular H₂. (d) eclogite sample DB11 (This study).

Samples DB 11 and DB 13 show weak peaks for any quantitative evaluation. Please any reasons for this?

33. These two samples (DB 11 and DB 13) are natural eclogite, and no special treatment has been carried out before measuring the water content. Therefore, the weak peaks in the infrared spectrum are due to their low water content (see Supplementary Table 6).

Tables

Extended Data Table 2

How is the modeled data correlating with the experimental data? What is T (temperature) values in this Table? Is it measured in K or °C?

34. We have added the fitting correlation (R^2) in Supplementary Table 2.

Unit of T (temperature) is in K. We have mentioned it in Supplementary Table 2 and text. (Line 100)

Extended Data Table 3

Apart from the samples IDs which belongs to the current work, and guess of geochronology by previous workers, the Table did not indicate which part belongs to the current research and which belongs to the previous work⁶⁰. It is also not clear whether the same samples were used by previous workers.

35. Supplementary Table 3 was reorganized to show it more clearly. As we mentioned in Line 242, the four samples used in this study were also used in Tang et al. (2007). The peak metamorphic conditions and age, protolith age, and exhumation age were determined by other workers (Zhang et al., 1995, 2009; Zheng et al., 2003; Liu et al., 2011). The mineral assemblage and major elements analyses were analyzed in this study.

Extended Data Table 4

No geochemical deductions were made from the major element compositions in this Table. Classification plots can be drawn from this data.

36. We have drawn a new figure (Supplementary Fig. 6) to show the classification for eclogites from the Dabie and Sulu terranes.

Supplementary Fig. 6 TAS (total alkali vs. silica) classification diagram for eclogites from the Dabie and Sulu terranes.

Extended Data Table 5

The water content after run on garnet for samples SL9, DB11 and DB13 have increased and SL12 has decreased. What could be causing this change? Why two samples from the same location have this disparity?

37. The water contents of garnet before experiments were 22, 101, 181, and 236 wt. ppm, after experiments were 42, 112, 170, 258 wt. ppm, respectively. However, the

general uncertainty of FTIR analysis is ~30% caused by a number of reasons including unpolarized infrared on unoriented minerals, baseline correction, sample thickness uncertainty, and so on. The variation is thus within the analytical uncertainty. We have added the explanation in Line 283-288.

Reviewer #3 (Comments to the Author):

This paper presents new experimental data on the thermal properties of natural eclogite, that are used to refine the thermal structure of orogenic crust in continental collision environments, and to discuss implications on partial melting (anatexis), which plays an important role on the rheology of the orogenic crust.

The new data are derived from rigorous and comprehensive experimental work, that consider the role of P–T, rock composition and H₂O content, and are placed within the framework of other data on eclogites. Furthermore, they provide the basis for numerical modeling to compare the thermal structure and anatectic potential of two orogenic systems with contrasting crustal thicknesses, the Dabie Shan/Sulu and the Tibetan plateau.

The results improve our understanding of thermal structure in orogenic belts, and inform on why orogenic continental crust undergoes partial melting. I focus the rest of my review in some aspects of concern. (# Refers to line numbers)

We thank the reviewer for the positive evaluation and helpful suggestions.

1- The manuscript gives the impression that partial melting of orogenic crust is still a controversial issue and that whether the continental crust can reach sufficiently high temperatures to produce partial melts, is questionable (ex. see #29–30, 54–55, 132–133). I agree that there is still a lot to be learned on this topic, but there are numerous petrological studies from crustal anatectic rocks from continental collision environments, and the authors mention the existence of independent evidence for melting in #130–131. In this context, this contribution is important not because it makes a case that anatexis can happen (something that we know anyways) but because it provides insight on why orogenic crust can partially melt.

1. Following the suggestions, we have focus more on why orogenic crust can partially melt, instead of focusing on whether the continental crust can melt or not. Thus, we have made changes accordingly. (Line 27, 39, 57 and 178)

2- There is some ambiguity in the use of certain terms and in some statements made. For instance:

(a) Continental collision results in crustal thickening and subduction of continental crust, but these two are not necessarily the same. This paper focuses on subduction, but crustal thickening and subduction are used interchangeably and this leads to some confusion.

2. Following the suggestions, we have revised the related sentences to avoid the misleading. “crustal thickening” and “subduction” are distinguished throughout the whole manuscript (for example Line 30-31, 50-51, 179).

(b) Eclogite is a major component of subduction zones (as stated in #49) and of deep levels of thickened continental crust, but it is not the dominant component of the thickened continental crust (as stated in #32)

3. We have corrected it., “we measured the thermal conductivity and diffusivity of natural eclogite, the most important ultrahigh pressure metamorphic rocks in subduction zones and in the deep part of thickened continental crust”. (Line 29-31)
“which is composed of omphacite and garnet and is the main constituent rock of subduction zones”. (Line 51-52)

3- Some critical information for the assessment of the manuscript is missing: this pertains to the age of high-P metamorphism and exhumation of the Dabie /Sulu belt, where the investigated samples come from. This is particularly important because the results are used to model and compare with the thermal structure of the Tibet, and because it is stated that the lithosphere in Dabie/Sulu there is thin (see #135–136). I guess this is a feature that developed later in the evolution of the belt?

4. The information on the peak metamorphic condition and age, protolith age, and exhumation age for the UHP eclogites from the Dabie-Sulu terrane were all added in Supplementary Table 3.

Right, the difference of lithospheric thickness between the Dabie-Sulu and Himalaya-Tibet belts is due to the later evolution of orogenic belt. In the revised manuscript, we have reorganized the discussion section on the partial melting and thermal structure of the lithospheric mantle following the evolution of the belts during subduction and exhumation. (Line 157-202)

Other

#34–37 30 km depth is close to the Moho discontinuity in ‘normal’, un-thickened continental crust; however, the calculations were done for thick, orogenic crust?

5. Our calculations were carried out for two typical orogenic crusts including Sulu belt with normal continental crust and Himalaya-Tibet belt with thickened crust. The related sentence is revised to make it clear (Line 28-39).

#47–48 ‘Early studies attributed these features to the hydration of eclogite...’ The reference in support to this (Liu et al., 2019) is a rather modern one...

6. “Early studies” is changed to “A few studies”. (Line 50)

#130–131 Felsic rocks also occur in these settings; they experience anatexis at lower pressures than mafic rocks, and their melting would also influence the rheology of the crust....

7. It is true that both mafic and felsic rocks experienced anatexis during exhumation of subducted continental crust. In the revised manuscript, we have reorganized it based on when and why orogenic continental crust undergoes partial melting in terms of the evolution of the belts. In Line 130-131, this sentence was rewritten as “Numerous studies have provided petrological evidences for the occurrence of partial melting in natural UHP rocks during the continental collision (Zeng et al., 2009; Zheng et al., 2011; Sawyer et al., 2013; Xu et al., 2013; Kohn, 2014; Wang et al., 2014; Li et al., 2016)”. (Line 141-144)

139–140 ‘..., the Moho temperature of the Sulu belt (~940 K) (Fig. 3a) is considerably lower than that of the Tibetan Plateau (~1340 K) (Fig. 3b), which implies a hotter Tibetan lithosphere.’ But also the Tibetan plateau Moho is much deeper than that under Sulu.... In fact for the same depth, Sulu lithosphere is hotter (see Fig. 3)

8. It’s true that Sulu and Tibetan plateau have different Moho depth.

To avoid any possible misleading, the related sentence is changed to “In the Sulu orogenic belt, the Moho temperature at depth of 32 km is ~940 K, and the inferred lithospheric thickness is around 64–67 km (Fig. 3a). In the Himalaya-Tibet orogenic belt, due to the differences in heat flow and heat production (Supplementary Fig. 5), the temperature at Moho discontinuity (~65 km) reaches to 1340 K, and the lithospheric thickness is close to 150 km (Fig. 3b).” (Line 158-162)

#148–156 This part is difficult to read without information about the evolution of the Dabie /Sulu belt, and on when was the thin lithosphere achieved (see also main point 3)

9. Based on the reviewer’s suggestion, this part was reorganized, with the discussion following the evolution of the belt. (Line 141-202)

#159–160 ‘the lowermost crust should be able to retain a protracted high temperature, as suggested by its low thermal diffusivity’ I don’t understand this statement; this doesn’t also depend upon the rate of tectonic processes?

10. It’s true that it also depends on the rate of tectonic processes.

The related sentence is deleted in the revised manuscript because we have reorganized it based on when and why orogenic continental crust undergoes partial melting in terms of the evolution of the belts. (Line 178-192)

Online content

#321 How is a ‘strong’ retrograde alteration is defined?

11. We mean the major minerals in eclogite are still garnet and omphacite. One sentence was added after Line 321 to make the definition more clear: “The concentrations of amphibole, epidote, biotite, aegirine, which are caused by garnet -> biotite + epidote, and omphacite -> amphibole + albite, are still very low (< 4 %).” (Line 244-246)

331 It would be very useful to add ages of peak metamorphism and exhumation in extended Table 3.

12. Age information was added in Supplementary Table 3.

REVIEWER COMMENTS

Reviewer #1 (Remarks to the Author):

The authors addressed all my comments. The addition of the explanation for the use of Zr really adds a lot to the paper, and the new results and figures are well detailed. They also provided interesting ideas moving forward to develop these membranes for higher performance. I agree with the authors that complete, or near complete, coverage could be possible. Challenging, but possible. Finally, all the materials and methods are now complete, which is good.

I do not have any further comments. I believe this is a very interesting work and I am looking forward to how these vertically aligned membranes will evolve.

Reviewer #2 (Remarks to the Author):

Although the authors provided responses to my previous comments, most of my major concerns, such as the lack of thorough material characterization data to sustain their claims, have not been well addressed. Moreover, there are two additional flaws in their revised manuscript:

1. Line 207 to 217, the authors claimed that "the free spacing of Zr-GO nanochannels can be calculated as 3.5 Å." However, such a claim is not consistent with their XRD data. In particular, XRD patterns (Supplementary Fig. 4b) indicate the d-spacing of 8.8 Å. After reducing the thickness of ~3.0 Å for one GO nanosheet, the free-spacing between the nanosheets (i.e., the channel height for mass transport) should be 5.8 Å, which is larger than the size of 2-propanol (~4.9 Å). In this case, 2-propanol vapor should transport through the GO film, which is contradictory to the authors' results (Figure 3b).
2. Even we assume their calculation was current and the free spacing is indeed 3.5 Å, such data still could explain their newly added liquid filtration data. The author claimed that "VAGME films show the expected high selectivity for Na(I) over the larger Mg(II) that is characteristic of GO nanochannel transport." The author further explained that: "The permeation rate of NaCl is two orders of magnitude higher than that of MgCl₂, which is consistent with the hydrated diameter difference between Na(I) and Mg(II)." These analyses indicate that their GO films exhibited much better rejection of NaCl than MgCl₂. Nevertheless, it should be emphasized that Na⁺ and Mg²⁺ ions have a hydration diameter of ~7.2 Å and 8.5 Å, respectively (Abraham, J. et al. Nat. Nanotechnol. 2017), such that both ions should be effectively rejected by their GO channels with the free spacing of 3.5 Å. Consequently, I would think there were some major flaws in their filtration test design or data interpretation. Additionally, the authors indicate that Na⁺ has a hydration diameter of 4.5 Å in Supplementary Table 4, which is another major mistake in the manuscript.

Again, these major concerns remarkably lower the reliability of their analysis and the overall quality of the study.

Response to reviewers' comments

Reviewer #2 (Remarks to the Author):

The revised manuscript has greatly improved and the authors have considered all the comments raised and effected all the corrections. The abstract is now written with no references. I recommend the acceptance of the manuscript with minor corrections.

Below are my edits:

Line 29 should read: the thermal conductivity and diffusivity of natural eclogite was measured instead of "we measured". The authors must avoid the use of personal pronouns in academic paper writing.

1. We agree in the main text we should avoid the personal pronouns. Therefore, all of such sentences in the main text are revised to passive voice.

But the abstract is an exception. Following the guidelines of Nature Communications (<https://www.nature.com/documents/nature-summary-paragraph.pdf>), we have to use "we measured..." in the abstract.

Line 57: As we know should be written as; "as it is known", or "from literature" instead of as we know.

2. "As we know" was replaced by "As it is known". (Line 50)

Similarly in line 77, avoid the use of we.

3. Following the suggestions, all the sentences with "we" in the main text are revised to passive voice. (Line 70, 97, 124, 140, 157, 166, 171-173)

Line 41 to 43: The later part of the abstract could be re-written for clarity.

4. The related text was rewritten as "The melt may facilitate the exhumation of subducted crust, produce the low-seismic-velocity zone, and cause the high-conductivity anomaly in the shallow depth of orogenic belts." (Line 33-35)

Reviewer #3 (Remarks to the Author):

The revised manuscript provides adequate clarifications on most points raised on the earlier version, and provides a solid contribution to the understanding of the thermal structure of orogenic crust based on thermal properties of eclogite. Still, the authors seem to link partial melting to subduction and to melting of eclogite, and this may be misleading because it is known that other rocks at shallower depths, outside of subduction zones, can also experience anatexis.

1. We totally agree that other rocks outside of subduction zones can also experience

anatexis. Our point is that, based on our results about eclogite thermal properties, temperature of the lithosphere is sufficient to melt fertile components in the deeper part of orogenic belts, therefore, answered the question about partial melting in orogenic continental crust (Line 32-33). We do not try to link partial melting in orogenic belts to the melting of eclogite. In this study, our purpose is to answer the questions about when and why partial melting have occurred in the continental crust of the Sulu and the Himalaya-Tibet belts during their subduction and exhumation. We are not focusing on other tectonic settings unrelated to subduction.

Detailed comments are listed below

#32-34 poorly phrased; specify that in Sulu the continental crust is restored to normal thickness whereas in Himalaya-Tibet it is still thick.

2. Due to the 150 words limit of the abstract, some descriptions about the Sulu and Himalaya-Tibet orogens have been moved to the Section of thermal structure simulation. (Line 141-142)

#35-39 In Sulu this thermal structure that led to the melting of eclogites was different than the present structure.

3. We do not mean melting of eclogite, but melting of fertile components (especially wet granite or phengite) (Line 32).

#37, 39 This study is concerned with melting of deep continental crust, and this has to be specified here; in orogenic systems, middle to lower crust can also melt in tectonic settings not related to subduction.

4. We agree with the reviewer's point. In Line 32, we pointed out that partial melting occurs in orogenic continental crust as long as the temperature exceeds the solidus temperature of fertile components (especially wet granite or phengite), whether it is a continental collision orogenic belt or outside of subduction zones. Please see also above Point 1.

#46-47 I agree that the low-velocity zones indicated here are consistent with the presence of melt, but, especially at shallower depths, this does not need to be derived from eclogite-facies rocks.

5. Thank you for your constructive comments. We do not mean that the melt at shallower depths was derived from eclogite-facies rocks, but due to the melting of fertile components (especially wet granite or phengite). (Line 161-165, 179-185, 190-195)

#146 olivine is not a rock, and amphibolite and granulite, but definitions, are not ultra-high-pressure rocks.

6. Following the suggestions, "ultrahigh-pressure rocks" was replaced by "rocks and minerals".